# Temporal Graph Benchmark
# for Machine Learning on Temporal Graphs

**Shenyang Huang**[1,2,*]   **Farimah Poursafaei**[1,2,*]   **Jacob Danovitch**[1,2]   **Matthias Fey**[3]
**Weihua Hu**[3]   **Emanuele Rossi**[4]   **Jure Leskovec**[7]   **Michael Bronstein**[8]
**Guillaume Rabusseau**[1,5,6]   **Reihaneh Rabbany**[1,2,6]

[1]Mila - Quebec AI Institute, [2]School of Computer Science, McGill University
[3]Kumo.AI, [4]Imperial College London, [5]DIRO, Université de Montréal
[6]CIFAR AI Chair, [7]Stanford University, [8]University of Oxford

## Abstract

We present the *Temporal Graph Benchmark (TGB)*, a collection of challenging and diverse benchmark datasets for realistic, reproducible, and robust evaluation of machine learning models on temporal graphs. TGB datasets are of large scale, spanning years in duration, incorporate both node and edge-level prediction tasks and cover a diverse set of domains including social, trade, transaction, and transportation networks. For both tasks, we design evaluation protocols based on realistic use-cases. We extensively benchmark each dataset and find that the performance of common models can vary drastically across datasets. In addition, on dynamic node property prediction tasks, we show that simple methods often achieve superior performance compared to existing temporal graph models. We believe that these findings open up opportunities for future research on temporal graphs. Finally, TGB provides an automated machine learning pipeline for reproducible and accessible temporal graph research, including data loading, experiment setup and performance evaluation. TGB will be maintained and updated on a regular basis and welcomes community feedback. TGB datasets, data loaders, example codes, evaluation setup, and leaderboards are publicly available at https://tgb.complexdatalab.com/.

## 1 Introduction

Many real-world systems such as social networks, transaction networks, and molecular structures can be effectively modeled as graphs, where nodes correspond to entities and edges are relations between entities. Recently, significant advances have been made for machine learning on static graphs, led by the use of *Graph Neural Networks (GNNs)* [22, 43, 7] and *Graph Transformers* [34, 23, 12], and accelerated by the availability of public datasets and standardized evaluations protocols, such as the widely adopted *Open Graph Benchmark (OGB)* [17].

However, most available graph datasets are designed only for *static* graphs and lack the fine-grained timestamp information often seen in many real-world networks that evolve over time. Examples include social networks [32], transportation networks [8], transaction networks [38] and trade networks [33]. Such networks are formalized as *Temporal Graphs (TGs)* where the nodes, edges, and their features change *dynamically*.

A variety of machine learning approaches tailored for learning on TGs have been proposed in recent years, often demonstrating promising performance [35, 9, 40, 25, 20]. However, Poursafaei *et al.* [33] recently revealed an important issue: these TG methods often portray an over-optimistic performance

---

*Equal contributions

— meaning they appear to perform better than they would in real-world applications — due to the inherent limitations of commonly used evaluation protocols.

This over-optimism creates serious challenges for researchers. It becomes increasingly difficult to distinguish between the strengths and weaknesses of various methods when their test results suggest similarly high performance. Furthermore, there is a discrepancy between real-world applications of TG methods and the existing evaluation protocols used to assess them. Therefore, there is a pressing need for an open and standardized benchmark that enhance the evaluation process for temporal graph learning, while being aligned with real-world applications.

In this work, we present the *Temporal Graph Benchmark (TGB)*, a collection of challenging and diverse benchmark datasets for realistic, reproducible, and robust evaluation for machine learning on temporal graphs. Figure 2 shows TGB's ML pipeline. Inspired by the success of OGB, TGB automates the process of dataset downloading and processing as well as evaluation protocols, and allows the user to easily compare their model performance with other models on the public leaderboard. TGB improves the evaluation of temporal graph learning in both *dataset selection* and *evaluation protocol* and covers both edge and node-level tasks.

**Dataset Selection.** Contrary to real-world networks that typically contain millions of nodes and tens of millions of edges, existing TG benchmark datasets are notably smaller, falling short by several orders of magnitude [33, 40]. Furthermore, these datasets often have limitations in terms of their domain diversity, with a substantial focus on social and interaction networks [35, 24, 25]. This lack of diversity can be problematic as network properties, such as network motifs [27], the scale-free property [4], and the modular structure [30] vary significantly across different domains. Consequently, it is important to benchmark existing methods across a wide variety of domains for a comprehensive evaluation.

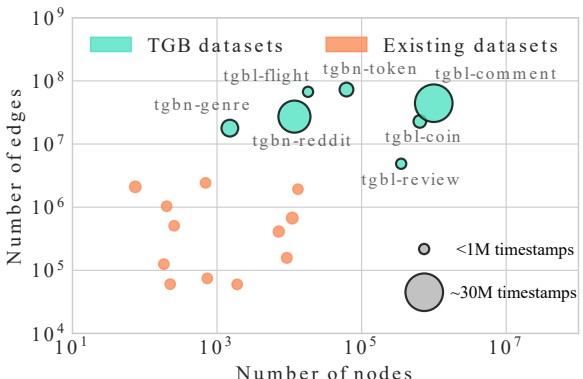

Figure 1: TGB consists of a diverse set of datasets that are one order of magnitude larger than existing datasets in terms of number of nodes, edges, and timestamps.

To address these limitations, TGB datasets provide diversity in terms of the number of nodes, edges, timestamps, and network domains. As shown in Figure 1, TGB datasets are larger in scale and present statistics that were under-explored in prior literature. For instance, the `tgbn-token` dataset has around 73 million edges while the `tgbl-comment` dataset has more than 30 million timestamps. Additionally, TGB introduces four datasets in the novel *node affinity prediction* task to address the scarcity of large scale datasets for node-level tasks in the current literature.

**Improved Evaluation.** In TGB, we aim to design the evaluation for both edge and node-level tasks on temporal graphs based on real applications. Historically, the standard approach for dynamic link prediction evaluation is to treat it as a binary classification task using one negative edge per positive edge in the test set [9, 40, 25, 33]. This strategy tends to generate negatives that are easy to predict, given the structure and sparsity of real-world networks [1], leading to inflated model performance estimations [33]. To address this issue, we propose to treat this task as a ranking problem, contrasting each positive sample against multiple negatives and using Mean Reciprocal Rank (MRR) as the metric. Moreover, *historical negatives* – past edges absent in the current step – are generally more difficult to predict correctly than randomly sampled negatives [33]. Thus, we sample both historical and random negatives in link prediction evaluations.

There is a lack of large-scale datasets for node-level tasks in temporal graphs as acquiring dynamic node labels remains challenging due to privacy concerns. We plan to include additional datasets for node classification in the future. As a starting point, we present the novel *node affinity prediction* task, which finds its motivation in recommendation systems. The objective is to anticipate the shifts in user preferences for items over time, as expanded in Section 3.2. In this task, node property is considered as the affinity towards different items at a given time. To assess the effectiveness of methods in addressing this task, we adopt the Normalized Discounted Cumulative Gain (NDCG) metric. This

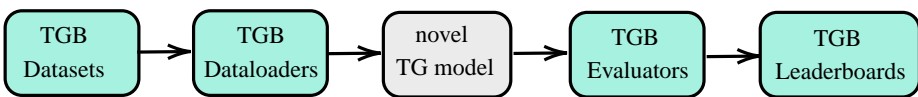

Figure 2: **Overview of the Temporal Graph Benchmark (TGB) pipeline: (a)** TGB includes large-scale and realistic datasets from five different domains with both dynamic link prediction and node property prediction tasks. **(b)** TGB automatically downloads datasets and processes them into `numpy`, `PyTorch` and `PyG` compatible `TemporalData` formats. **(c)** Novel TG models can be easily evaluated on TGB datasets via reproducible and realistic evaluation protocols. **(d)** TGB provides public and online leaderboards to track recent developments in temporal graph learning domain. The code is publicly available as a Python library.

metric serves to determine whether the methods' predictions for class importance adhere to the same ordering as the ground truth. In Section 5.2, we show that simple heuristics can outperform state-of-the-art TG methods in achieving superior performance for this task.

Overall, our proposed Temporal Graph Benchmark has the following contributions:

- **Large and diverse datasets.** TGB includes datasets coming from a diverse range of domains and spanning both edge and node-level tasks. We contributed seven novel datasets which are orders of magnitude larger than existing ones in terms of number of edges, nodes and timestamps.

- **Improved evaluation.** We propose an improved and standardized evaluation pipeline motivated by real-world applications. For dynamic link property prediction, we sample multiple negative instances per positive edge and ensure a mix of both historical and random negative samples, while using the *MRR* metric. For the node affinity prediction task, we use the *NDCG* metric to evaluate the relative importance of classes within the top ranking ones.

- **Empirical findings.** We show that for the dynamic link property prediction task, model performances can vary drastically across datasets. For example, the best performing model on `tgbl-wiki` encounters a 40% test MRR drop on `tgbl-review` and the *surprise index* of a dataset affects model performance. On the node affinity prediction task, we find that simple heuristics can often outperform state-of-the-art TG methods, thus leaving ample room for development of future methods targeting this task.

- **Public leaderboard and reproducible results.** Following the good practice of OGB, TGB also provides an automated and reproducible pipeline for both link and node property prediction tasks. Researchers can submit and compare method performance on the TGB leaderboard.

**Reproducibility:** TGB code, datasets, leaderboards and details are on the TGB website. The code is also publicly available on GitHub with documentations seen here.

## 2 Related Work

**Temporal Graph Datasets and Libraries.** Recently, Poursafaei *et al.* [33] collected six novel datasets for link prediction on continuous-time dynamic graphs while proposing more difficult negative samplings for evaluation. In comparison, we curated seven novel temporal graph datasets spanning both edge and node-level tasks for realistic evaluation of machine learning on temporal graphs. Yu *et al.* [52] presented DyGLib, a platform for reproducible training and evaluation of existing TG models on common benchmark datasets. DyGLib demonstrates the discrepancy of the model performance across different datasets and argues that diverse evaluation protocols of previous works caused an inconsistency in performance reports. Similarly, Skarding *et al.* [39] provided a comprehensive comparative analysis of heuristics, static GNNs, discrete dynamic GNN, and continuous dynamic GNN on dynamic link prediction task. They showed that dynamic models outperforms their static counterparts consistently and heuristic approaches can achieve strong performance. In all of the above benchmarks, the included datasets only contain a few million edges. In comparison, TGB datasets are orders of magnitude larger in scale in terms of number of nodes, edges and timestamps. TGB also includes both node and edge-level tasks. Huang *et al.* [18] collected a novel dynamic

graph dataset for anomalous node detection in financial networks and compared the performance of different graph anomaly detection methods. In this work, TGB datasets have more edges and timestamps while covering both edge and node tasks.

**Temporal Graph Methods.** With the growing interest in temporal graph learning, several recent models achieved outstanding performance on existing benchmark datasets. However, due to the limitations of the current evaluation, many methods achieve over-optimistic and similar performance for the dynamic link prediction task [46, 35, 9, 40, 25, 24]. In this work, TGB datasets and evaluation show a clear distinction between SOTA model performance, which helps facilitate future advancement of TG learning methods. Temporal graphs are categorized into discrete-time and continuous-time temporal graphs [21]. In this work, we focus on the continuous-time temporal graphs as it is more general. Continuous-time TG methods can be divided into node or edge representation learning methods. Node-based models such as *TGN* [35], *DyRep* [42] and *TCL* [45] first leverage the node information such as temporal neighborhood or previous node history to generate node embeddings and then aggregate node embeddings from both source and destination node of an edge to predict its existence. In comparison, edge-based methods such as *CAWN* [46] and *GraphMixer* [9] aim to directly generate embeddings for the edge of interest and then predict its existence. Lastly, the simple memory-based heuristic *EdgeBank* [33] without any learning component has shown surprising performance based on existing evaluation. We compare these methods on TGB datasets in Section 5. For more discussion on TG methods see Appendix D.

## 3   Task Evaluation on Temporal Graphs

Temporal graphs are often used to model networks that evolve over time where nodes are entities and temporal edges are relations between entities through time. In this work, we focus on continuous-time temporal graphs and denote them as timestamped edge streams consisting of triplets of source, destination, and timestamp; i.e., $\mathcal{G} = \{(s_0, d_0, t_0), (s_1, d_1, t_1), \ldots, (s_T, d_T, t_T)\}$ where the timestamps are ordered ($0 \leq t_1 \leq t_2 \leq \ldots \leq t_T$) [33, 21]. Note that temporal graph edges can have different properties namely being weighted, directed, or attributed. We consider $\mathcal{G}_t$ as the augmented graph of all edges observed in the stream up to the time $t$ with nodes as $\mathbf{V}_t$ and edges as $\mathbf{E}_t$. Optionally, $\mathcal{G}_t$ can contain node features $\mathbf{X}_t \in \mathbb{R}^{|\mathbf{V}_t| \times k_n}$ where $k_n$ is the size of a node feature vector, and edge features $\mathbf{M}_t \in \mathbb{R}^{|\mathbf{E}_t| \times k_m}$ where $k_m$ is the size of an edge feature vector. We consider a fixed chronological split to form the training, validation, and test set.

**Evaluation Settings.** There are several possible evaluation settings in the temporal graph based on the available information of the test set. We categorize and discuss these settings in detail in Appendix C. In this work, we consider the *streaming setting* where the deployed models need to adapt to new information at inference time. More specifically, we follow the setting in [35] where previously observed test edges can be accessed by the model but back-propagation and weight updates with the test information are not permitted.

### 3.1   Dynamic Link Property Prediction

The goal of *dynamic link property prediction* is to predict the property (oftentimes the existence) of a link between a node pair at a future timestamp. The timeline is chronologically split at two fixed points resulting in three sets of edges $E_{\text{train}}$, $E_{\text{val.}}$, and $E_{\text{test}}$. In TGB, we improve the evaluation setting in the following ways.

**Negative edge sampling.** In current evaluation [24, 48, 35, 45, 40, 51, 6, 53], only one negative edge is sampled uniformly randomly from all possible node pairs to evaluate against each positive edge. In contrast, in real applications where the true edges are not known in advance, the edges with the highest probabilities predicted by a given model are used to decide which connections should be prioritized. With that in mind, we treat the link prediction task as a ranking problem and sample multiple negative edges per each positive edge. In particular, for a given positive edge $e^p : (s, d, t)$, we fix the source node $s$ and timestamp $t$, and sample $q$ different destination nodes. For each dataset, we select the number of negative edges $q$ based on the trade-off between evaluation completeness and the test set inference time.

We sample the negative edges from both the *historical* and *random* negative edges. Historical negatives are sampled from the set of edges that are observed in the training set but are not present at the current timestamp $t$ (*i.e.* $E_t \setminus E_{\text{train}}$), they are shown to be more difficult for models to predict

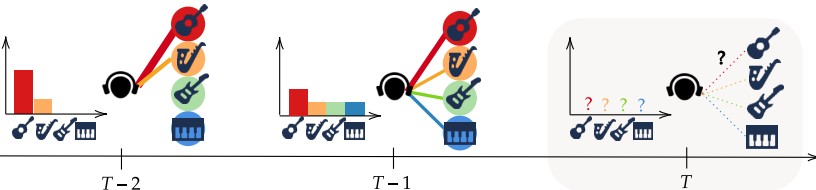

Figure 3: The *node affinity prediction* task aims to predict how the preference of a user towards items changes over time. In the `tgbn-genre` example, the task is to predict the frequency at which the user would listen to each genre over the next week given their listening history until today.

than random negatives [33]. We sample equally from historical and random negative edges. Note that depending on the dataset and the timestamp $t$, there might not be enough historical negatives to sample from. In this case, we simply increase the ratio of the random negatives to have the desired number of negative edges per positive ones. For reproducibility, we include a fixed set of negatives sampled for each dataset to ensure consistent comparison amongst models.

**Performance metric.** The commonly used metric for reporting models' performance for the dynamic link prediction task is either Area Under the Receiver Operating Characteristic curve (AUROC) or Average Precision (AP). An appropriate metric should be able to capture the ranking of a positive edge amongst the negative ones, which is not fulfilled by either AUROC or AP. Thus, we devise to use the filtered Mean Reciprocal Rank (MRR) as the evaluation metric for the dynamic link property prediction. The MRR computes the reciprocal rank of the true destination node among the negative or fake destinations. The MRR varies in the range of $(0, 1]$ and it is a commonly used metric in recommendation systems [47] and knowledge graphs [44, 17, 16]. In addition, recent link prediction literature is also shifting towards adopting the MRR metric [5, 16, 50]. It should be noted that when reporting the MRR, we perform collision checks to ensure that no positive edge is sampled as a negative edge.

## 3.2 Dynamic Node Property Prediction

The goal of *dynamic node property prediction* is to predict the property of a node at any given timestamp $t$, i.e., to learn a function $f : \mathbf{V}_t \rightarrow \mathcal{Y}$, where $\mathbf{V}_t$ is the set of nodes at time $t$ and $\mathcal{Y}$ is some output space (e.g. $\{-1, +1\}, \mathbb{R}, \mathbb{R}^p$, etc.). This is a general category for node-level tasks such as node classification and node regression. Here, the property of the node can be a one hot label, a weight vector or an euclidean coordinate. Currently, there is a lack of large-scale temporal graph datasets with node labels in the literature; therefore, we first include the *node affinity prediction* task (as defined below). In the future, we will add more node-level tasks and datasets into TGB.

**Node affinity prediction.** This task considers the affinity of a subset of nodes (representing, e.g., users) towards other nodes (e.g., items) as its property, and how the affinity naturally changes over time. This task is relevant for example in recommendation systems, where it is important to provide personalized recommendations for a user by modelling their preference towards different items over time. Figure 3 shows the node affinity prediction task in the context of music recommendation systems as seen in the `tgbn-genre` dataset. In this task, we are given the interaction history of a user with different music genres, and the goal is to predict the frequency at which the user would listen to each genre over the next week.

More formally, given the observed evolution history of a temporal graph $\mathcal{G}_t$ until current timestamp $t$, the *node affinity prediction* task (on a dataset such as `tgbn-genre`) predicts the interaction frequency vector $\mathbf{y}_t[u, :]$ for a node $u$ over a set of candidate nodes $\mathbf{N}$ within a fixed future period $[t, t+k]$ where $k$ is the window size defined by the application. Each entry in $\mathbf{y}_t[u, :]$ corresponds to a candidate node $v \in \mathbf{N}$ and the groundtruth value is generated as follows:

$$\mathbf{y}_t[u, v] = \frac{\sum_{t < t_i \leq t+k} w_{(u,v,t_i)}}{\sum_{z \in \mathbf{N}} \sum_{t < t_i \leq t+k} w_{(u,z,t_i)}} \tag{1}$$

where $w_{(u,v,t_i)}$ is the weight of the edge $(u, v, t_i)$ (which we assume to be 0 if the edge between $u$ and $v$ is not present at time $t_i$). Observe that, by definition, $\|\mathbf{y}_t[u, :]\|_1 = 1$. We use the Normalized Discounted Cumulative Gain (NDCG) metric that takes into account the relative order of elements.

Table 1: Dataset Statistics. Dataset names are colored based on their scale as small, medium, and large. ¶: Edges can be *Weighted*, *Directed*, or *Attributed*.

| | Dataset | Domain | # Nodes | # Edges | # Steps | Surprise | Edge Properties¶ |
|---|---|---|---|---|---|---|---|
| **Link** | tgbl-wiki | interact. | 9,227 | 157,474 | 152,757 | 0.108 | W: ✗, Di: ✓, A: ✓ |
| | tgbl-review | rating | 352,637 | 4,873,540 | 6,865 | 0.987 | W: ✓, Di: ✓, A: ✗ |
| | tgbl-coin | transact. | 638,486 | 22,809,486 | 1,295,720 | 0.120 | W: ✓, Di: ✓, A: ✗ |
| | tgbl-comment | social | 994,790 | 44,314,507 | 30,998,030 | 0.823 | W: ✓, Di: ✓, A: ✓ |
| | tgbl-flight | traffic | 18143 | 67,169,570 | 1,385 | 0.024 | W: ✗, Di: ✓, A: ✓ |
| **Node** | tgbn-trade | trade | 255 | 468,245 | 32 | 0.023 | W: ✓, Di: ✓, A: ✗ |
| | tgbn-genre | interact. | 1,505 | 17,858,395 | 133,758 | 0.005 | W: ✓, Di: ✓, A: ✗ |
| | tgbn-reddit | social | 11,766 | 27,174,118 | 21,889,537 | 0.013 | W: ✓, Di: ✓, A: ✗ |
| | tgbn-token | transact. | 61,756 | 72,936,998 | 2,036,524 | 0.014 | W: ✓, Di: ✓, A: ✓ |

NDCG is commonly used in information retrieval and recommendation systems as a measure of ranking quality [19]. In this work, we use NDCG@10 where the relative order of the top 10 ranked items (*i.e.,* destination nodes) are examined. Specifically in the tgbn-genre dataset, the NDCG@10 compares the ground truth to the relative order of the top-10 music genres that a model predicts.

## 4 Datasets

TGB offers nine temporal graph datasets, seven of which are collected and curated for this work. All datasets are split chronologically into the training, validation, and test sets, respectively containing $70\%, 15\%$, and $15\%$ of all edges, in line with similar studies such as [48, 35, 45, 20, 40, 25]. The dataset licenses and download links are presented in Appendix B, and the datasets will be permanently maintained via Digital Research Alliance of Canada funded by the Government of Canada. We consider datasets with more than 5 million edges as medium-size and those with more than 25 million edges as large-size datasets.

Table 1 shows the statistics and properties of the temporal graph datasets provided by TGB. Datasets such as tgbl-flight, tgbl-comment, tgbl-coin, and tgbn-reddit are orders of magnitude larger than existing TG benchmark datasets [35, 33, 24], while their number of nodes and edges span a wide spectrum, ranging from thousands to millions. In addition, TGB dataset domains are highly diverse, coming from five distinct domains including social networks, interaction networks, rating networks, traffic networks, and trade networks. Moreover, the duration of the datasets varies from months to years, and the number of timestamps in TGB datasets ranges from 32 to more than 30 million with diverse ranges of time granularity from UNIX timestamps to annually. The datasets can be weighted, directed, or have edge attributes. We also report the *surprise index* (i.e., $\frac{|E_{test} \setminus E_{train}|}{|E_{test}|}$) as defined in [33] which computes the ratio of test edges that are not seen during training. Low surprise index implies that memorization-based methods (such as EdgeBank [33]) can potentially achieve good performance on dynamic link property prediction task. We can observe that the surprise index also varies notably across TGB datasets, further contributing to datasets diversity. We present more dataset statistics in Appendix G. We discuss the details of TGB datasets next.

tgbl-wiki. This dataset stores the co-editing network on Wikipedia pages over one month. The network is a bipartite interaction network where editors and wiki pages are nodes, while one edge represents a given user edits a page at a specific timestamp. Each edge has text features from the page edits. The task for this dataset is to predict with which wiki page a user will interact at a given time.

tgbl-review. This dataset is an Amazon product review network from 1997 to 2018 where users rate different products in the electronics category from a scale of one to five. Therefore, the network is a bipartite weighted network where both users and products are nodes and each edge represents a particular review from a user to a product at a given time. Only users with a minimum of 10 reviews within the aforementioned time interval are kept in the network. The considered task for this dataset is to predict which product a user will review at a given time.

tgbl-coin. This is a cryptocurrency transaction dataset based on the Stablecoin ERC20 transactions dataset [37]. Each node is an address and each edge represents the transfer of funds from one address

to another at a time. The network starts from April 1st, 2022, and ends on November 1st, 2022, and contains transaction data of 5 stablecoins and 1 wrapped token. This duration includes the Terra Luna crash where the token lost its fixed price of 1 USD. The considered task for this dataset is to predict with which destination a given address will interact at a given time.

`tgbl-comment.` This dataset is a directed reply network of Reddit where users reply to each other's threads. Each node is a user and each interaction is a reply from one user to another. The network starts from 2005 and ends at 2010. The considered task for this dataset is to predict if a given user will reply to another one at a given time.

`tgbl-flight.` This dataset is a crowd sourced international flight network from 2019 to 2022. The airports are modeled as nodes, while the edges are flights between airports at a given day. The node features include the type of the airport, the continent where the airport is located, the ISO region code of the airport as well as its longitude and latitude. The edge feature is the associated flight number. In this dataset, our task is to predict whether a flight will happen between two specific airport on a future date. This is useful for foreseeing potential flight disruptions such as cancellation and delays. For instance, during the COVID-19 pandemic, many flight routes were cancelled to combat the spread of COVID-19. In addition, the prediction of global flight network is also important for studying and forecasting the spread of disease such as COVID-19 to new regions, as shown in [3, 11].

`tgbn-trade.` This is the international agriculture trading network between nations of the United Nations (UN) from 1986 to 2016. Each node is a nation and an edge represents the sum trade value of all agriculture products from one nation to another one. As the data is reported annually, the time granularity of the dataset is yearly. The considered task for this dataset is to predict the proportion of agriculture trade values from one nation to other nations during the next year.

`tgbn-genre.` This is a bipartite and weighted interaction network between users and the music genres of songs they listen to. Both users and music genres are represented as nodes while an interaction specifies a user listens to a music genre at a given time. The edge weights denote the percentage of which a song belongs to a certain genre. The dataset is constructed by cross referencing the songs in the *LastFM-song-listens* dataset [24, 15] with that of music genres in the *million-song* dataset [2]. The *LastFM-song-listens* dataset has one month of who-listens-to-which-song information for 1000 users and the *million-song* dataset provides genre weights for all songs in the *LastFM-song-listens* dataset. We only retain genres with at least 10% weights for each song that are repeated at least a thousand times in the dataset. Genre names are cleaned to remove typos. Here, the task is to predict how frequently each user will interact with music genres over the next week. This is applicable to many music recommendation systems where providing personalized recommendation is important and user preference shifts over time.

`tgbn-reddit.` This is a users and subreddits interaction network. Both users and subreddits are nodes and each edge indicates that a user posted on a subreddit at a given time. The dataset spans from 2008 to 2010. The task considered for this dataset is to learn the interaction frequency towards the subreddits of a user over the next week.

`tgbn-token.` This is a user and cryptocurrency token transaction network. Both users and tokens are nodes and each edge indicates the transaction from a user to a token. The edge weights indicate the amount of token transferred and considering the disparity between weights, we normalized the edge weights using logarithm. The goal here is to predict how frequently a user will interact with various types of tokens over the next week. The dataset is extracted and curated from this source [37].

## 5 Experiments

For dynamic link property prediction, we include DyRep [42], TGN [35], CAWN [46], TCL [45], GraphMixer [9], NAT [25], TGAT [48] and two deterministic heuristics namely EdgeBank$_{tw}$ and EdgeBank$_\infty$ [33]. For dynamic node property prediction, we include DyRep, TGN, and deterministic heuristics such as persistence forecast [36] and moving average [31]. Details about the above methods are presented in Appendix D. The computing resources are discussed Appendix E. For the experimental results, we report the average and standard deviation across 5 different runs. We highlight the best results in bold and underline the second place results.

Table 2: Results for *dynamic link property prediction* on small datasets.

| Method | MRR | |
|---|---|---|
| | Validation | Test |
| DyRep [42] | $0.072_{\pm 0.009}$ | $0.050_{\pm 0.017}$ |
| TGN [35] | $0.435_{\pm 0.069}$ | $0.396_{\pm 0.060}$ |
| CAWN [46] | $\underline{0.743}_{\pm 0.004}$ | $\underline{0.711}_{\pm 0.006}$ |
| TCL [45] | $0.198_{\pm 0.016}$ | $0.207_{\pm 0.025}$ |
| GraphMixer [9] | $0.113_{\pm 0.003}$ | $0.118_{\pm 0.002}$ |
| TGAT [48] | $0.131_{\pm 0.008}$ | $0.141_{\pm 0.007}$ |
| NAT [25] | $\mathbf{0.773}_{\pm 0.011}$ | $\mathbf{0.749}_{\pm 0.010}$ |
| EdgeBank$_{tw}$ [33] | 0.600 | 0.571 |
| EdgeBank$_{\infty}$ [33] | 0.527 | 0.495 |

(a) `tgbl-wiki` dataset with *surprise index*: 0.108 (with *all* negative edges per each positive edge).

| Method | MRR | |
|---|---|---|
| | Validation | Test |
| DyRep [42] | $0.216_{\pm 0.031}$ | $0.220_{\pm 0.030}$ |
| TGN [35] | $0.313_{\pm 0.012}$ | $0.349_{\pm 0.020}$ |
| CAWN[46] | $0.200_{\pm 0.001}$ | $0.193_{\pm 0.001}$ |
| TCL [45] | $0.199_{\pm 0.007}$ | $0.193_{\pm 0.009}$ |
| GraphMixer [9] | $\mathbf{0.428}_{\pm 0.019}$ | $\mathbf{0.521}_{\pm 0.015}$ |
| TGAT [48] | $\underline{0.324}_{\pm 0.006}$ | $\underline{0.355}_{\pm 0.012}$ |
| NAT [25] | $0.302_{\pm 0.011}$ | $0.341_{\pm 0.020}$ |
| EdgeBank$_{tw}$ [33] | 0.0242 | 0.0253 |
| EdgeBank$_{\infty}$ [33] | 0.0229 | 0.0229 |

(b) `tgbl-review` dataset with *surprise index*: 0.987 (with *100* negative edges per each positive edge).

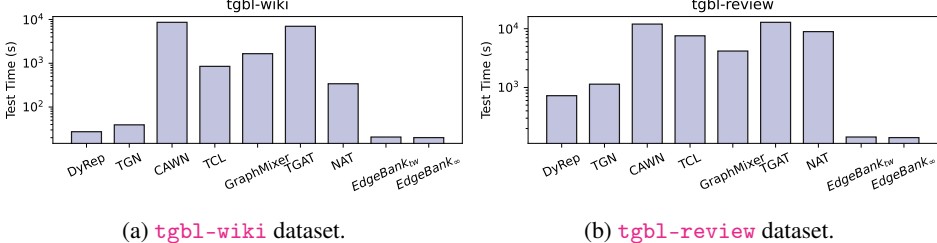

(a) `tgbl-wiki` dataset.      (b) `tgbl-review` dataset.

Figure 4: Test set inference time of TG methods can have up to two orders of magnitude difference.

## 5.1 Dynamic Link Property Prediction

Table 2a shows the performance of TG methods for dynamic link property prediction on the `tgbl-wiki` dataset. `tgbl-wiki` is an existing dataset where many methods achieve over-optimistic performance in the literature [35, 46, 9]. With TGB's evaluation protocol, there is now a clear distinction in model performance and NAT achieves the best result on this dataset. As `tgbl-wiki` is the smallest dataset in this task, it is computationally feasible to sample all possible destinations of a given source node. Thus, we compare the true destination with all possible negative destinations in this dataset. In Table 2b, we report the results on `tgbl-review` where we sample 100 negative edges per positive edge. Here, we observe that many of the best performing methods on `tgbl-wiki` has a significant drop in performance including NAT, CAWN and Edgebank. More notably, the method rankings also changed significantly with GraphMixer and TGAT being the top two methods. This observation emphasizes the importance of dataset diversity when benchmarking TG methods. In Appendix H, we conduct an ablation study on the effect of number of negative samples on the performance of dynamic link property prediction.

One explanation for the significant performance reduction of some methods on `tgbl-review` is that it has a higher *surprise index* compared to `tgbl-wiki` (see Table 1). The surprise index reflects the ratio of edges in the test set that have not been seen during training. Therefore, a dataset with a high surprise index requires more inductive reasoning, as most of the test edges are unobserved during training. As a heuristic that memorizes past edges, EdgeBank performance is inversely correlated with the surprise index and it achieves higher performance when the suprise index of the dataset is low. An interesting future direction is the investigation of the performance of certain category of methods with the surprise index. For example, the top two methods NAT and CAWN on wikipedia both utilizes features from the joint neighborhood of nodes in the queried edge [25]. On the `tgbl-review` dataset, the best performing TGAT is designed for inductive representation learning on temporal graph [48] which fits the inductive nature of `tgbl-review` which has high surprise index.

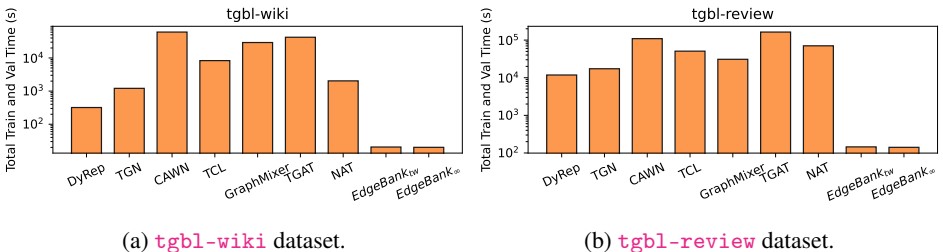

(a) `tgbl-wiki` dataset.     (b) `tgbl-review` dataset.

Figure 5: Total train and validation time of TG methods can have two orders of magnitude difference.

| Method | tgbl-coin MRR | | tgbl-comment MRR | | tgbl-flight MRR | |
|---|---|---|---|---|---|---|
| | Validation | Test | Validation | Test | Validation | Test |
| DyRep [42] | $0.512_{\pm 0.014}$ | $0.452_{\pm 0.046}$ | $0.291_{\pm 0.028}$ | $0.289_{\pm 0.033}$ | $0.573_{\pm 0.013}$ | $0.556_{\pm 0.014}$ |
| TGN [35] | $\mathbf{0.607}_{\pm 0.014}$ | $\mathbf{0.586}_{\pm 0.037}$ | $\mathbf{0.356}_{\pm 0.019}$ | $\mathbf{0.379}_{\pm 0.021}$ | $\mathbf{0.731}_{\pm 0.010}$ | $\mathbf{0.705}_{\pm 0.020}$ |
| EdgeBank$_{tw}$ [33] | 0.492 | 0.580 | 0.124 | 0.149 | 0.363 | 0.387 |
| EdgeBank$_{\infty}$ [33] | 0.315 | 0.359 | 0.109 | 0.129 | 0.166 | 0.167 |

Table 3: Results for *dynamic link property prediction* task on medium and large datasets.

Figure 4a and 4b show the inference time of different methods for the test set of `tgbl-wiki` and `tgbl-review`, respectively. Similarly, Figure 5a and 5b show the total training and validation time of TG methods for `tgbl-wiki` and `tgbl-review`. Notice that as a heuristic baseline, EdgeBank inference, train, or validation time is generally at least one order of magnitude lower than neural network based methods. We also observe an order of difference in inference time within TG methods. We believe one important future direction is to improve the computational time of these models to be closer to baselines such as EdgeBank, which can better scale to large real-world temporal graphs.

Table 3 shows the performance of TG methods on medium and large TGB datasets. Note that some methods, including CAWN, TCL, and GraphMixer, ran out of memory on GPU for these datasets, thus their performance is not reported. Overall, TGN has the best performance on all of these three datasets. Surprisingly, the EdgeBank heuristic is highly competitive on the `tgbl-coin` dataset where it even significantly outperforms DyRep. Therefore, it is important to include EdgeBank as a baseline for all datasets. Another observation is that for medium and large TGB datasets, there can be a significant performance change for a single model between the validation and test set. This is because TGB datasets span over a long time (such as `tgbl-comment`, lasting 5 years) and one can expect that models need to deal with potential distribution shifts between the validation set and the test set. Figure 6a, 6b and 6c reports the test time for TG methods on `tgbl-coin`, `tgbl-flight` and `tgbl-comment`, respectively. On both `tgbl-coin` and `tgbl-comment`, Edgebank is at least one order of magnitude faster than TGN and DyRep while on the `tgbl-flight`, due to the large number of temporal edges, DyRep is the fastest method.

## 5.2 Dynamic Node Property Prediction

Table 4 shows the performance of various methods on the node affinity prediction task in the dynamic node property prediction category. As node-level tasks have received less attention compared to edge-level tasks in the literature, adopting methods that are specially designed for link prediction to this task is non-trivial. As a result, these methods are omitted in this section. Considering Table 4, the key observation is that simple heuristics like persistence forecast and moving average are strong contenders to TG methods such as DyRep and TGN. Notably, persistence forecast is SOTA on `tgbn-trade` while moving average is the best performing on other datasets. TGN is second place on `tgbn-genre` dataset. Different from link prediction where the existence of a link is casted as binary classification, the node affinity prediction task compares the likelihood or weight that the model assigns to different target nodes (mostly positive links). These results highlight the need for the future development of TG methods that can acquire flexible node representations capable of learning how user preferences evolve over time.

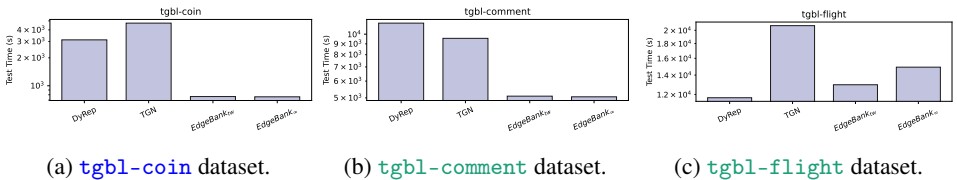

| (a) `tgbl-coin` dataset. | (b) `tgbl-comment` dataset. | (c) `tgbl-flight` dataset. |

Figure 6: Inference time comparison of TG methods.

| Method | tgbn-trade NDCG@10 | | tgbn-genre NDCG@10 | | tgbn-reddit NDCG@10 | | tgbn-token NDCG@10 | |
|---|---|---|---|---|---|---|---|---|
| | Validation | Test | Validation | Test | Validation | Test | Validation | Test |
| DyRep [42] | $0.394_{\pm0.001}$ | $0.374_{\pm0.001}$ | $0.357_{\pm0.001}$ | $0.351_{\pm0.001}$ | $0.344_{\pm0.001}$ | $0.312_{\pm0.001}$ | $0.151_{\pm0.006}$ | $0.141_{\pm0.006}$ |
| TGN [35] | $0.395_{\pm0.002}$ | $0.374_{\pm0.001}$ | $0.403_{\pm0.010}$ | $0.367_{\pm0.058}$ | $0.379_{\pm0.004}$ | $0.315_{\pm0.020}$ | $0.189_{\pm0.005}$ | $0.169_{\pm0.006}$ |
| persistence Fore. [36] | **0.860** | **0.855** | 0.350 | 0.357 | 0.380 | 0.369 | 0.403 | 0.430 |
| Moving Avg. [31] | 0.841 | 0.823 | **0.499** | **0.509** | **0.574** | **0.559** | **0.491** | **0.508** |

Table 4: *Node affinity prediction* results.

# 6 Conclusion

To enable realistic, reproducible, and robust evaluation for machine learning on temporal graphs, we present the Temporal Graph Benchmark, a collection of challenging and diverse datasets. TGB datasets are diverse in their dataset properties as well as being orders of magnitude larger than existing ones. TGB includes both *dynamic link property prediction* and *dynamic node property prediction* tasks, while providing an automated pipeline for researchers to evaluate novel methods and compare them on the TGB leaderboards. In dynamic link property prediction, we find that model rankings can vary significantly across datasets, thus demonstrating the necessity to evaluate on the diverse range of TGB datasets. Surprisingly for dynamic node property prediction, simple heuristics such as persistence forecast and moving average outperforms SOTA methods such as TGN. This motivates the development of more TG methods for node-level tasks.

**Impact on Temporal Graph Learning.** Significant advancements in machine learning are often accelerated by the availability of public and well-curated datasets such as ImageNet [10] and OGB [17]. We expect TGB to be a common and standard benchmark for temporal graph learning, helping to facilitate novel methodological changes.

**Potential Negative Impact.** If TGB becomes a widely-used benchmark for temporal graph learning, it is possible that future papers might focus on TGB datasets and tasks, which may limit the use of other TG tasks and datasets for benchmarking. To avoid this issue, we plan to update TGB regularly with community feedback as well as adding additional datasets and tasks.

**Limitations.** Firstly, TGB only considers the most common TG evaluation setting as discussed in Section 3, namely the *streaming* setting. We also discuss other possible settings in Appendix C. Depending on a specific application, a different setting might be more suitable (such as forbidding test time node updates). Secondly, TGB currently only contains datasets from five domains, while many other domains such as biological networks are not included. We plan to continue adding datasets to TGB to further increase the dataset diversity in TGB.

## Acknowledgments and Disclosure of Funding

We thank the OGB team for sharing their website theme in the construction of this project's website. We also thank Elahe Kooshafar for giving advice on the collection of the `tgbn-genre` dataset. This research was supported by the Canadian Institute for Advanced Research (CIFAR AI chair program), Natural Sciences and Engineering Research Council of Canada (NSERC) Postgraduate Scholarship-Doctoral (PGS D) Award and Fonds de recherche du Québec - Nature et Technologies (FRQNT) Doctoral Award. We would like to thank Samsung Electronics Co., Ldt. for partially funding this research. Michael Bronstein and Emanuele Rossi are supported in part by ERC Consolidator Grant No. 274228 (LEMAN).

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

## A  Dataset Documentation and Intended Use

All datasets presented by TGB are intended for academic use and their corresponding licenses are listed in Appendix B. For the ease of access, we provide the following links to the TGB benchmark suits and datasets.

- Dataset and project documentations can be found at: `https://tgb.complexdatalab.com/`.

- TGB `Python` package is available via `pypi` at: `https://pypi.org/project/py-tgb/`.

- Tutorials and API references can be found at: `https://docs.tgb.complexdatalab.com/`.

**Maintenance Plan.** To provide a robust, realistic, and reproducible benchmark for temporal graphs, we plan to continue developing and maintaining TGB based on community feedback and involvement. We will maintain and improve the TGB and TGB-Baselines github repository, while the TGB datasets are maintained via Digital Research Alliance of Canada (funded by the Government of Canada).

## B  Dataset Licenses and Download Links

In this section, we present dataset licenses and the download link (embedded in dataset name). The datasets are maintained via Digital Reseach Alliance of Canada funded by the Government of Canada. As authors, we confirm the data licenses as indicated below and that we bear all responsibility in case of violation of rights.

- `tgbl-wiki`: MIT license. The original dataset can be found here [24].

- `tgbl-review`: Amazon license. By accessing the Amazon Customer Reviews Library (a.k.a. *Reviews Library*), one agrees that the Reviews Library is an Amazon Service subject to the *Amazon.com* Conditions of Use and one agrees to be bound by them, with the following additional conditions. In addition to the license rights granted under the Conditions of Use, Amazon or its content providers grant the user a limited, non-exclusive, non-transferable, non-sublicensable, revocable license to access and use the Reviews Library for purposes of academic research. One may not resell, republish, or make any commercial use of the Reviews Library or its contents, including use of the Reviews Library for commercial research, such as research related to a funding or consultancy contract, internship, or other relationship in which the results are provided for a fee or delivered to a for-profit organization. One may not (a) link or associate content in the Reviews Library with any personal information (including Amazon customer accounts) or (b) attempt to determine the identity of the author of any content in the Reviews Library. If one violates any of the foregoing conditions, their license to access and use the Reviews Library will automatically terminate without prejudice to any of the other rights or remedies Amazon may have. The original dataset can be found here [29].

- `tgbl-coin`: CC BY-NC license (Attribution-NonCommercial). The original dataset can be found here [37].

- `tgbl-comment`: CC BY-NC license (Attribution-NonCommercial). The original dataset can be found here [28].

- `tgbl-flight`: a non-commercial, limited, non-exclusive, non-transferable, non-assignable, and terminable license to copy, modify, and use the data in accordance with this agreement solely for the purpose of non-profit research, non-profit education, commercial internal testing and evaluation of the data, or for government purposes. No license is granted for any other purpose and there are no implied licenses in this agreement. For more details, please consult the original dataset license. The original dataset can be found here [41].

- `tgbn-trade`: MIT license. The original dataset can be found here [26].

- `tgbn-genre`: MIT license. The original *LastFM-song-listens* dataset [24, 15] and the *million-song* dataset [2] are available here and here, respectively.

- `tgbn-reddit`: CC BY-NC license (Attribution-NonCommercial). The original dataset can be found here [28].

- `tgbn-token`: CC BY-NC license (Attribution-NonCommercial). The original dataset can be found here [37].

# C   Evaluation Settings in TG

Based on how the test set information is used during the evaluation of temporal graph learning models, the existing approaches can be grouped into the following three categories. We emphasize that methods from different categories should be evaluated distinctly to avoid unfair comparison. Following the practice of the previous works and for the sake of clarity, we comply with *streaming setting*. Particularly, we consider a fixed split point in time, i.e. $t_{split}$, where the information before and after this point constitute the training and test data, respectively.

**Streaming Setting.** In this setting, information from the test set is only employed for updating the *memory* module in temporal graph learning methods (e.g., TGN [35]). However, no back-propagation or model update is possible based on the test set information. We consider this setting as *streaming*, since it helps in fast inference by incorporating the recent information (even from the test set), while observign the fact that retraining the model with the test data is too expensive.

**Deployed Setting.** In this setting, the test set information is not available for any modification to the model. This setting closely follows the standard machine learning setting with distinct training dataset and test dataset. We refer to this setting as *deployed* to denote that after deploying the a model, it is only used for inference and no updates to any part of the model is allowed.

**Live-Update Setting.** In this setting, information from any point in the past (including the test set information) can be used to re-train, fine-tune, or update the model. The goal of this setting is to achieve the best prediction for each timestamp $t + 1$ given the historical information at all previous timestamps in $[0, ..., t]$. We consider this setting as *live-update* because the model weights can be updated lively through the incoming data. Note that this setting is similar to the rolling setting exercised in ROLAND framework [51].

# D   Temporal Graph Learning Models

Recently, there is an increasing interest in developing graph representation learning models for networks that evolve over time. Evolving graphs can be investigated at different time granularity. Kazemi *et al.* [21] categorize temporal graphs into Discrete Time Dynamic Graphs (DTDGs) and Continous Time Dynamic Graphs (CTDGs). DTDGs consist of an ordered set of static graph snapshots while CTDGs compose of timestamped edge streams where edges are denoted by triplets of source, destination, and timestamp. DTDG models specialize in sequences of graph snapshots capturing the dynamics of a temporal graph where these snapshots are sampled at regular time intervals [51, 13, 32, 14, 49]. In contrast, CTDG methods often aggregate the features from temporal neighborhood (connected nodes with close time proximity) and constantly adjust the encoding as edge stream continues [35, 46, 25]. In this work, we primarily focus on CTDGs, driven by several compelling reasons. First, CTDG models which employ the precise temporal information offered by CTDGs, can be effectively utilized for DTDGs. On the contrary, adapting DTDG models for drawing inferences from CTDGs is a non-trivial task. Second, it has been shown that DTDGs are convertible to CTDGs [40], while the conversion in the opposite direction results in loss of information. Next, we introduce the methods used in our evaluation for the dynamic link property prediction and dynamic node property prediction task.

**Models Used for Evaluation of Dynamic Link Property Prediction Task.**

- *DyRep* [42] is a temporal point process-based model that propagates interaction messages via Recurrent Neural Networks (RNNs) to update the node representations. It employs a temporal attention mechanism to model the weights of a given node's neighbors.

- *TGN* [35] is a general framework for learning on continuous time dynamic graphs. It has the following components: memory module, message function, message aggregator, memory updater, and embedding module. TGN updates the node memories at test time with newly observed edges.

- *GraphMixer* [9] is a simple model for dynamic link prediction consisting of three main modules: a node-encoder to summarize the node information, a link-encoder to summarize the temporal link information, and a link predictor module. These modules that only utilize Multi-Layer Perceptrons (MLPs), make GraphMixer a simple model without the use of a GNN architecture.

- *TCL* [45] employs a transfomer module to generate temporal neighborhood representations for nodes involved in an interaction. It then models the inter-dependencies with a co-attentional trans-

fomer at a semantic level. Specifically, TCL utilizes two separate encoders to extract representations from temporal neighborhoods surrounding the two nodes involved in an edge.

- *CAWN* [46] predicts dynamic links based on extracting temporal random walks and retrieves temporal network motifs to represent network dynamics. It utilizes a neural network model to encode Causal Anonymous Walks (CAWs) to support online training and inference. Particularly, it starts by relabeling nodes with encoded temporal anonymous walks starting at the nodes involved in an interaction. Then, the temporal walks themselves are further encoded using the generated node labels and encodings of the elapsed times. Finally, the existence of a link is predicted based on aggregating the walks encoding through a pooling module.

- *TGAT* [48] captures both temporal and structural characteristics of dynamic graphs using a self-attention mechanism. At a specific timestamp $t$, TGAT first concatenates the raw features of a node with a trainable encoding of time $t$. Upon the observation of an interaction among a pair of nodes, TGAT utilizes the self-attention mechanism to apply message passing to the time augmented node features and generates the latest node representations.

- *NAT* [25] adopts a novel dictionary-type neighborhood representation for nodes. Such representation records a downsampled set of node neighbors as keys and constructs features for joint neighborhood of multiple nodes. A dedicated data structure called N-cache supports GPU acceleration for the dictionary representations.

- *EdgeBank$_\infty$* [33] is a simple heuristic storing all observed edges in a memory (implemented as a hashtable). At inference time, if the queried node pair is in the memory then EdgeBank$_\infty$ predicts true, otherwise EdgeBank$_\infty$ predicts false.

- *EdgeBank$_{tw}$* [33] is another variation of EdgeBank which only memorizes edges from a fixed duration in the recent past. Therefore, it has a strong recency bias.

**Models Used for Evaluation of Dynamic Node Property Prediction Task.**

- *TGN* [35] is discussed above. We utilize the TGN embedding of a node from the memory at the query time $t$ to predict the node labels.

- *DyRep* [42] same as discussed above. We use the node embedding to predict the node labels.

- *Persistant Forecast* [36] is a simple yet powerful baseline for time series forecasting and complex systems. Here, we extend the core idea by simply outputting the recently observed node label for the current time $t$.

- *Moving Average* [31] considers the average of the node labels observed in the previous $k$ steps (we set $k = 7$).

## E    Computing Resources

**Dynamic link property prediction.** For this task, we ran all experiments on either Narval or Béluga cluster of Digital Research Alliance of Canada. For the experiments on Narval cluster, we ran each experiment on a Nvidia A100 (40G memory) GPU with 4 CPU nodes (from either of the AMD Rome 7532 @ 2.40 GHz 256M cache L3, AMD Rome 7502 @ 2.50 GHz 128M cache L3, or AMD Milan 7413 @ 2.65 GHz 128M cache L3 available type) each with 100G memory. For the experiments on Béluga, we ran each experiment on a NVidia V100SXM2 (16G memory) GPU with 4 CPU nodes (from either of the Intel Gold 6148 Skylake @ 2.4 GHz, Intel Gold 6148 Skylake @ 2.4 GHz, Intel Gold 6148 Skylake @ 2.4 GHz, or Intel Gold 6148 Skylake @ 2.4 GHz) each with 100G memory. A five-day time limit is considered for each experiment. We repeated each experiments five times and reported the average and standard deviation of different runs. It is noteworthy that except for the TGN [35] and DyRep [42] models that we ported them into the `PyTorch Geometric` environment, the other models (evaluated by their original source code or by using the DyGLib repository) throw an *out of memory* error for the medium and large datasets on both Narval and Béluga clusters.

**Dynamic Node Property Prediction** For this task, we ran experiments with 4 standard CPU and either RTX8000, V100, A100 or A6000 GPUs. The longest experiment takes around 2 days on the `tgbl-comment` and `tgbn-token` dataset.

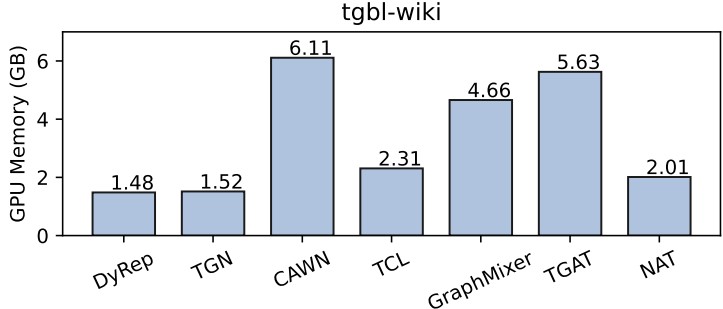

Figure 7: GPU Memory Usage for `tgbl-wiki` dataset.

Table 5: Additional dataset properties. Dataset names are colored based on their scale as small, medium, and large. ⋆: denotes the average number of edges per timestamp.

| | Dataset | inductive node ratio | | # inductive nodes | | total # nodes | | $\|\overline{E_t}\|^\star$ | Reoccurrence |
|---|---|---|---|---|---|---|---|---|---|
| | | Val. | Test | Val. | Test | Val. | Test | | |
| Link | tgbl-wiki | 0.257 | 0.308 | 836 | 1,099 | 3,256 | 3,564 | 1.031 | 0.015 |
| | tgbl-review | 0.024 | 0.027 | 5,728 | 6,336 | 242,820 | 234,641 | 709.911 | 0.0002 |
| | tgbl-coin | 0.112 | 0.174 | 37,689 | 336,161 | 55,781 | 321,008 | 17.604 | 0.025 |
| | tgbl-comment | 0.474 | 0.562 | 137,424 | 177,558 | 289,713 | 315,662 | 1.430 | 0.006 |
| | tgbl-flight | 0.031 | 0.045 | 474 | 689 | 15,170 | 15,347 | 48497.884 | 0.009 |
| Node | tgbn-trade | 0.009 | 0.009 | 2 | 2 | 216 | 228 | 15104.677 | 0.052 |
| | tgbn-genre | 0.056 | 0.097 | 70 | 129 | 1,260 | 1,328 | 4.265 | 0.003 |
| | tgbn-reddit | 0.031 | 0.035 | 349 | 390 | 11,191 | 11,099 | 1.241 | 0.009 |
| | tgbn-token | 0.086 | 0.076 | 4,013 | 3,178 | 46,541 | 42,023 | 35.81 | 0.002 |

## F  GPU Usage Comparison

In Figure 7, we report the average GPU usage of TG methods on the `tgbl-wiki` dataset across 5 trials. Note that EdgeBank is a heuristic and only requires CPU thus no GPU usage is reported. Some methods such as GraphMixer has significantly higher GPU usage when compared to others while most methods have similar GPU usage.

## G  Additional Dataset Statistics

In addition to the main dataset statistics presented in Table 1, it is insightful to examine some other dataset characteristics as indicated in Table 5. The *reoccurrence index* (i.e., $\frac{|E_{train} \cap E_{test}|}{|E_{train}|}$) denotes the ratio of training edges that reoccur during the test phase as well. If the edges appearance follows a consistent pattern, a high *reoccurrence index* can be correlated with the high performance of a memorization-based approach such as EdgeBank [33]. The average number of edges per timestamps provides information about the evolution of the datasets, and the ratio of the new nodes in validation or test set provides insights about the portion of unseen nodes introduced during inference. It should be noted that although we mainly focus on *transductive* dynamic link property prediction task in our evaluation, the ratio of new nodes in validation or test set (fourth and fifth column in Table 5) show that there are indeed new nodes during the validation and test phase. Correctly predicting the properties of edges for the new nodes might be more challenging for the models, since no historical information about these nodes are available. Lastly, we observe that TGB datasets are also diverse in all of the above properties.

## H  Number of Negative Samples

To study the effect of the number of negative samples, here we report the results for the small link property prediction datasets in TGB with *20* negative samples in Table 6a and 6b for `tgbl-wiki` and

Table 6: Results for *dynamic link property prediction* on small datasets with *20* negative samples.

| Method | MRR | |
|---|---|---|
| | Validation | Test |
| DyRep [42] | $0.411_{\pm 0.015}$ | $0.366_{\pm 0.014}$ |
| TGN [35] | $\underline{0.737}_{\pm 0.004}$ | $\underline{0.721}_{\pm 0.004}$ |
| CAWN [46] | $\mathbf{0.794}_{\pm 0.014}$ | $\mathbf{0.791}_{\pm 0.015}$ |
| TCL [45] | $0.734_{\pm 0.007}$ | $0.712_{\pm 0.007}$ |
| GraphMixer [9] | $0.707_{\pm 0.014}$ | $0.701_{\pm 0.014}$ |
| EdgeBank$_{tw}$ [33] | 0.641 | 0.641 |
| EdgeBank$_{\infty}$ [33] | 0.551 | 0.538 |

(a) `tgbl-wiki` dataset with *surprise index*: 0.108. (with *20* negative edges per positive edge).

| Method | MRR | |
|---|---|---|
| | Validation | Test |
| DyRep [42] | $0.356_{\pm 0.016}$ | $0.367_{\pm 0.013}$ |
| TGN [35] | $\mathbf{0.465}_{\pm 0.010}$ | $\mathbf{0.532}_{\pm 0.020}$ |
| CAWN[46] | $0.201_{\pm 0.002}$ | $0.194_{\pm 0.004}$ |
| TCL [45] | $0.194_{\pm 0.012}$ | $0.200_{\pm 0.010}$ |
| GraphMixer [9] | $\underline{0.411}_{\pm 0.025}$ | $\underline{0.514}_{\pm 0.020}$ |
| EdgeBank$_{tw}$ [33] | 0.0894 | 0.0836 |
| EdgeBank$_{\infty}$ [33] | 0.0786 | 0.0795 |

(b) `tgbl-review` dataset with *surprise index*: 0.987. (with *20* negatives edges per positive edge).

Table 7: Transductive vs. Inductive Setting on `tgbl-wiki` Dataset.

| Method | Val. MRR | | Test MRR | |
|---|---|---|---|---|
| | Transductive | Inductive | Transductive | Inductive |
| DyRep [42] | $0.058_{\pm 0.013}$ | $0.049_{\pm 0.021}$ | $0.040_{\pm 0.017}$ | $0.053_{\pm 0.021}$ |
| TGN [35] | $0.395_{\pm 0.028}$ | $0.231_{\pm 0.028}$ | $0.325_{\pm 0.055}$ | $0.268_{\pm 0.043}$ |
| TCL [45] | $0.187_{\pm 0.012}$ | $0.212_{\pm 0.016}$ | $0.194_{\pm 0.014}$ | $0.182_{\pm 0.009}$ |
| GraphMixer [9] | $0.113_{\pm 0.005}$ | $0.130_{\pm 0.004}$ | $0.122_{\pm 0.005}$ | $0.107_{\pm 0.005}$ |
| NAT [25] | $\mathbf{0.784}_{\pm 0.012}$ | $\mathbf{0.745}_{\pm 0.002}$ | $\mathbf{0.763}_{\pm 0.011}$ | $\mathbf{0.717}_{\pm 0.005}$ |
| CAWN [46] | $\underline{0.749}_{\pm 0.009}$ | $\underline{0.742}_{\pm 0.004}$ | $\underline{0.720}_{\pm 0.010}$ | $\underline{0.709}_{\pm 0.005}$ |
| TGAT [48] | $0.147_{\pm 0.009}$ | $0.150_{\pm 0.020}$ | $0.155_{\pm 0.007}$ | $0.152_{\pm 0.016}$ |
| EdgeBank$_{tw}$ [33] | 0.605 | 0.566 | 0.575 | 0.551 |
| EdgeBank$_{\infty}$ [33] | 0.520 | 0.566 | 0.481 | 0.562 |

`tgbl-review` across five trials, respectively. When contrasting these findings with those presented in Table 2a and 2b, which involve a higher quantity of negative samples, the insights outlined in Section 5 are reaffirmed. For example, there is a significant drop in performance for the top performing method, i.e. CAWN, from `tgbl-wiki` to `tgbl-review`. In addition, Edgebank performance has a significant drop when the surprise index is high (which is the case in `tgbl-review`). However, we can also clearly see that with lower number of negatives (particularly on `tgbl-wiki`), the MRR scores are significantly higher for most methods. Therefore, if computationally feasible, it is best to use more negative samples. In our case, more negative samples are used for small TGB datasets.

## I  Transductive and Inductive Comparison

In this section, we compare method performance following the commonly used transductive and inductive setting as seen in prior literature [48, 46] across five trials. We define inductive nodes as nodes that are unseen in the training set and transduction nodes as nodes that are observed in the training set. The transductive MRR is computed from all edges which has a transdustive source node and inductive MRR is computed from all edges that has an inductive source node. We observe that for a method to achieve strong performance on `tgbl-wiki` such as NAT, it should perform well in both transductive and inductive reasoning.

