# OpenReview forum: "Temporal Graph Benchmark for Machine Learning on Temporal Graphs"
_NeurIPS.cc/2023/Track/Datasets_and_Benchmarks — NeurIPS 2023 Datasets and Benchmarks Poster_

### Official Review · Reviewer_12NE · 2023-07-18
**New Temporal Graph Benchmark with new findings and potential future research directions**

**Rating:** 9
**Confidence:** 5
**Clarity:** This paper is well-written.

**Strengths:**

S1. This study provides new findings on the characteristics of GNNs and suggests future research directions as follows.
1) The definition of the "surprise index" (the proportion of new links that have never occurred in the past) and the analysis of its correlation with GNN design is intriguing. Particularly, this study shows that CAWN achieves high accuracy (low accuracy) when the surprise index is low (high). Additionally, it is revealed that EdgeBank, a memorization-based method, performs well for the datasets with a low surprise index (predicting existing links) but struggles in the opposite case.
2) The evaluation results of inference time suggest scalability as a future research direction, as the heuristic baseline, EdgeBank, is an order of magnitude faster than other advanced methods.
3) As for the dynamic node property prediction task, this study shows potential for further research because different methods achieve the highest accuracy on different datasets.

S2. This benchmark covers various scale and diverse application domains datasets (beyond typical social and interaction domains) and has been publicly released including datasets and leaderboards. Six out of the eight datasets are newly released.

S3. This benchmark utilizes two types of negative sampling (typical random samples and historical negatives) and employs practical metrics (MRR for link prediction and NDCG for node property prediction) instead of using AUC.

S4. This benchmark supports both link prediction and dynamic node property prediction tasks, making it comprehensive and versatile.

**Additional Feedback:**

Not anymore.

**Correctness:**

Yes, the benchmark design is appropriate and the evaluations were performed correctly.

**Documentation:**

The detailed information is included for the datasets and the benchmark is already publically available.

**Ethics:**

No ethical concerns.

**Limitations:**

The limitations are adequately addressed in Section 6.

**Opportunities For Improvement:**

O1. The benchmarks should include not only inference time evaluation but also training time evaluation.

O2. It is desirable to add some latest methods such as NAT and NeurTWs.

O3. The publication year of [25] is not mentioned.

O4. It would be better to describe the fixed future period [t, t+k] for the link prediction task too, if its design is similar to the case of the dynamic node property prediction task.

**Relation To Prior Work:**

Yes, the new contributions are clear.

**Summary And Contributions:**

This study proposes Temporal Graph Benchmark (TGB), a new benchmark for machine learning on temporal graphs by following the same concept as Open Graph Benchmark (OGB) designed for static graphs. The contributions are as follows:
1. The datasets cover various scale and application domains. Also, it has already been publicly released including the datasets and leaderboards.
2. The benchmark incorporates both difficult historical negatives and simple random negatives, and utilizes ranking metrics (MRR for link prediction and NDCG for node property prediction) which are more practical than AUC.
3. The experiments reveal new findings on the characteristics of GNNs and suggest future research directions (See S1 in the Strengths).

---

> ### Author Response · Authors · 2023-08-17
> **Author Response to Reviewer 12NE**
>
> We thank the reviewer for their time, valuable feedback and the appreciation of our work.
>
> **O1: the benchmarks should include not only inference time evaluation but also training time evaluation.**
>
> **A:** Thanks for this suggestion. We have added *Figure 5* which shows the comparison of the total training and validation time for all methods on tgbl-wiki and tgbl-review. Additional time comparisons on other datasets can be found in *Appendix J*. We have the same observation that there exist orders of magnitudes gap in compute time between TG methods thus scalability research is an important future research direction.
>
> **O2: It is desirable to add some latest methods such as NAT and NeurTWs.**
>
> **A:** We have added NAT to the comparison in *Section 5.1*. We have reached out to the authors of NeurTWs to ask if they would like to submit their method to the TGB leaderboard. We refer the reviewer to our joint response to all reviewers for discussions on additional baselines.
>
> **O3: The publication year of [25] is not mentioned.**
>
> **A:** Thanks for pointing this out. We have added the publication year to the reference in the revision.
>
> **O4: It would be better to describe the fixed future period [t, t+k] for the link prediction task too, if its design is similar to the case of the dynamic node property prediction task.**
>
> **A:** For the link property prediction task, we use the same set up as the standard dynamic link prediction task where the goal is to predict if a link exists at time *t* from source node *u* to destination node *v*. Currently, the fixed future period is only used in the node property prediction task.

---

### Official Review · Reviewer_87xo · 2023-07-20
**Reviews of 120**

**Rating:** 7
**Confidence:** 4
**Correctness:** Yes
**Clarity:** Yes

**Strengths:**

1 The authors have made a clear contribution to the dynamic graph learning community by providing a comprehensive and systematic benchmark.

2 The scale and richness of the authors' dataset is commendable.

**Additional Feedback:**

The current score is provisional, hope to see the author's good reply.

**Documentation:**

Yes

**Limitations:**

Yes

**Opportunities For Improvement:**

1 Figure 3 is a little difficult to understand, I suggest adding more annotations and instructions to the figure to help readers understand.

2 Did the author consider node classification or clustering on the dynamic graph in the task setting? These are two important tasks in graph learning, but they are rarely mentioned in this article. Does this mean that this article does not focus on the classification of nodes?

3 Does the author know about the DGraph data set? As far as I know, this is also a larger data set on dynamic graphs, focusing on issues such as anomaly detection on dynamic graphs. Did the authors consider comparisons with such datasets? In other words, whether in terms of scale or scope of application, what advantages does this paper have over DGraph?
[1] DGraph: A Large-Scale Financial Dataset for Graph Anomaly Detection. NeurIPS 2022.

**Relation To Prior Work:**

Yes

**Summary And Contributions:**

The authors present the Temporal Graph Benchmark (TGB), a collection of challenging and diverse benchmark datasets for realistic, reproducible, and robust evaluation of machine learning models on temporal graphs.

---

> ### Author Response · Authors · 2023-08-17
> **Author Response to Reviewer 87xo**
>
> We thank the reviewer for their time and valuable feedback.
>
> **Q1: Figure 3 is a little difficult to understand**
>
> **A:** We have updated *Figure 3* in the revision and added more clarification in the caption as well as discussions in Section 3.2. We hope this makes the Figure more clear and we welcome any additional feedback to improve it.
>
> **Q2: Did the author consider node classification or clustering on the dynamic graph in the task setting?**
>
> **A:** We have considered the node classification task initially but were not able to find any large scale dataset for this task. The current node classification benchmark datasets used in TG literature [1] (mostly wikipedia and reddit) are small in scale (less than two million edges). We plan to add node classification / node clustering datasets to TGB in the near future, including the DGraph dataset kindly suggested by the reviewer.
>
> [1] E. Rossi, B. Chamberlain, F. Frasca, D. Eynard, F. Monti, and M. Bronstein. Temporal graph networks for deep learning on dynamic graphs. arXiv preprint arXiv:2006.10637, 2020.
>
> **Q3: Does the author know about the DGraph data set? In other words, whether in terms of scale or scope of application, what advantages does this paper have over DGraph?**
>
> **A:** We thank the reviewer for pointing out this reference, we were not aware of this dataset before. DGraph dataset is indeed an interesting dataset with labels for anomaly detection on dynamic graphs. In terms of scale, the DGraph dataset has around 4.3 million edges, 3.7 million nodes and 821 timestamps. In comparison, TGB has nine datasets with up to 73 million edges, up to 1 million nodes and up to 30 million timestamps. Thus, TGB is larger in number of edges and timestamps while DGraph has more nodes. In terms of applications, TGB has both edge and node level tasks which are important for applications such as recommendation systems and traffic forecasting while DGraph focuses on anomaly detection in financial networks. Given the large number of labels in DGraph, we plan to incorporate it into TGB in the near future. We also added discussion of DGraph to *Section 2* in the revision.

---

> > ### Comment · Reviewer_87xo · 2023-08-26
> >
> > Thanks for the author's response, I will consider raising the rating.

---

### Official Review · Reviewer_WDxN · 2023-07-21
**Review for Submission 120**

**Rating:** 6
**Confidence:** 5
**Correctness:** Yes.
**Clarity:** The paper is well written.

**Strengths:**

1. TGB provides diverse datasets and a pipeline for both link and node property prediction tasks on temporal graphs. TGB uses novel metrics (MRR and NDCG) for temporal graph tasks. TGB solves an important issue: existing TG methods often portray an over-optimistic performance due to the inherent limitations of commonly used evaluation protocols.
2. TGB code, datasets, leaderboards and details are on the TGB website (https://tgbwebsite.pages.dev/).
3. This research can be considered relevant to the broader research community because all datasets and codes for temporal graph tasks are publicly accessible under license.
4. For link prediction task, this paper conducts experiences on five benchmark datasets (MRR metric) with seven TG models (DyRep, TGN, CAWN, TCL, GraphMixer, EdgeBanktw, and EdgeBank). For node prediction task, this paper conducts experiences on three benchmark datasets (NDCG metric) with four TG models (DyRep, TGN, Persistent Fore., and Moving Avg.).

**Additional Feedback:**

None.

**Documentation:**

Yes.

**Ethics:**

No.

**Limitations:**

Yes. The authors have adequately addressed the limitations and potential negative societal impact of their work in Section 6.

**Opportunities For Improvement:**

1. In TGs (DyRep, TGN, and CAWN, etc.), AUC (Area Under the ROC Curve) and AP (Average Precision) are the common metrics for the link and node prediction tasks. However, in this research, MRR and NDCG metrics are used as improved and standardized evaluation protocols. Thus, it is difficult to compare experimental results with previous studies. AUC and AP metrics should be considered too.
2. For the Dynamic Link Property Prediction task, TGB treats the link prediction task as a ranking problem and samples multiple negative edges per each positive edge. TGB samples the negative edges from both the historical and random negative edges. However, in section 5.1, there are no experimental results to illustrate the difference between historical and random negative edges. Generally, the transductive and inductive settings for the link prediction task should be considered too.
3. For the Dynamic Link Property Prediction task, some state-of-the-art TG models should be
considered. For example:
Neural Temporal Walks: Motif-Aware Representation Learning on Continuous-Time Dynamic Graphs. NeurIPS 2022
Neighborhood-aware Scalable Temporal Network Representation Learning. LoG 2022
Provably expressive temporal graph networks. NeurIPS 2022
4. It woule be better to add more datasets for the link and node prediction tasks. For example, the datasets (MOOC, LastFM, Enron, SocialEvo, US Legis., UN Trade, etc.) in EdgeBank.
5. More efficiency comparison of TG models should be considered (not only test set inference time in Figure 4), including training time, memory usage, etc.

**Relation To Prior Work:**

Yes. In Section 1 and Section 2, this research clearly discussed how this work differs from previous works.

**Summary And Contributions:**

This paper presents Temporal Graph Benchmark (TGB), a collection of challenging and diverse benchmark datasets for evaluation of machine learning models on temporal graphs (TGs) where the nodes, edges, and their features change dynamically. TGB provides large and diverse datasets and an automated machine learning pipeline for reproducible and accessible temporal graph research, including data loading, experiment setup and performance evaluation.

---

> ### Author Response · Authors · 2023-08-17
> **Author Response to Reviewer WDxN**
>
> We thank the reviewer for their time and valuable feedback.
>
> **Q1: AUC and AP metrics should be considered too.**
>
> **A:** In previous work where AUC and AP metrics are used, the evaluation consists of discriminating between one positive edge and one randomly sampled negative edge (as seen in TGN, CAWN and DyRep etc.). However, this leads to inflated and over-optimistic model performance. Here, we improve the evaluation by comparing a positive edge with multiple negatives (which is better aligned with practical applications of these methods). We chose MRR as the metric for link property prediction because it captures the ranking of a positive edge among many negatives and is widely used in recommendation systems and knowledge graph literature. In addition, recent link prediction literature [1,2] is shifting towards reporting MRR. See more discussions in the revised *Section 3.1*.
>
> [1] W. Hu, M. Fey, H. Ren, M. Nakata, Y. Dong, and J. Leskovec. Ogb-lsc: A large-scale challenge for machine learning on graphs. arXiv preprint arXiv:2103.09430, 2021.
>
> [2] B. P. Chamberlain, S. Shirobokov, E. Rossi, F. Frasca, T. Markovich, N. Y. Hammerla, M. M.Bronstein, and M. Hansmire. Graph neural networks for link prediction with subgraph sketching.In The Eleventh International Conference on Learning Representations, 2022.
>
> **Q2: there are no experimental results to illustrate the difference between historical and random negative edges.**
>
> **A:** In [1], the authors provided a detailed experimental analysis between historical and random negative edges when one positive edge is evaluated against one negative edge, showing that historical negatives are indeed more difficult than random negative edges. We added a mention in the revised *Section 3.1*.
>
> [1] F. Poursafaei, A. Huang, K. Pelrine, and R. Rabbany. Towards better evaluation for dynamic link prediction. In Thirty-sixth Conference on Neural Information Processing Systems Datasets and Benchmarks Track.
>
> **Q3: Generally, the transductive and inductive settings for the link prediction task should be considered too**
>
> **A:** We refer the reviewer to our joint response to all reviewers for a detailed discussion. We have added an additional experiment of transductive and inductive setting in *Appendix I*.In the following table (for test set of tgbl-wiki), we show the comparison between the two settings. We also showed the amount of inductive nodes in TGB datasets in *Table 5* and believe that strong performance for both transductive and inductive nodes are required to achieve high score on the TGB leaderboard.
>
> | Method  |   Transductive MRR |  Inductive MRR |
> |----------|:-------------:|:-------------:|
> | DyRep |   0.040   |   0.053   |
> | TGN   |   0.325   |   0.268    |
> | CAWN  |  0.720   |  0.709     |
> | TCL    |  0.194   |  0.182     |
> | GraphMixer |  0.122    |  0.107     |
> | **TGAT**  |  0.155    |  0.152     |
> | **NAT**   |  **0.763**    |  **0.717**   |
> | $EdgeBank_{tw}$     |  0.575    |  0.551     |
> | $EdgeBank_{\infty}$     |  0.481    |  0.562    |
>
>
>
> **Q4: some state-of-the-art TG models should be considered**
>
> **A:** We have added additional baselines including NAT and TGAT to the revised *Section 5.1*. Please refer to our joint response to all reviewers for more details. We have reached out to the author of NeurTWs to ask if they would like to submit their method to the TGB leaderboard. We have started running PINT on tgbl-wiki and tgbl-review. For tgbl-wiki, PINT hasn’t finished running after 4 days while on tgbl-review, it is out of memory. We reached out to the authors to ask if they would like to submit to TGB leaderboard.
>
> **Q5:  add more datasets for the link and node prediction tasks**
>
> **A:** We plan to continue to add new datasets to TGB based on community feedback. In this work, we focus on large scale datasets which are closer to the scale seen in real-world applications. In this revision, we have added one novel dataset: tgbn-token, a dataset with 73 million edges for node property prediction (see results below and in *Section 5.2*). The datasets from EdgeBank are diverse but limited in scale (< 3 million edges) thus not considered to be added at the moment.
>
> | Method  |  Val NDCG |  Test NDCG |
> |----------|:-------------:|:-------------:|
> | DyRep |   0.151   |   0.141 |
> | TGN |   0.189   |   0.169 |
> | Persistence Fore.  |  0.403   |  0.430 |
> | Moving Avg. |  **0.491**   |  **0.508** |
>
>
> **Q6: More efficiency comparison of TG models should be considered**
>
> **A:** We thank the reviewer for this suggestion. We have added *Figure 5* to *Section 5.1* which reports the total training and validation time of all methods on tgbl-wiki and tgbl-review dataset. In *Appendix J*, we report additional efficiency comparison for other datasets. Lastly, in *Appendix F* (*Figure 6*), we added the GPU memory usage comparison between all methods. From all efficiency comparisons, we observe that scalability is an important future research direction.

---

> > ### Comment · Reviewer_WDxN · 2023-08-28
> > **Response**
> >
> > Thanks for the feedback.
> >
> > The authors have addressed most of my concerns. I have increased my score accordingly.

---

### Official Review · Reviewer_yfpt · 2023-07-21
**Review on "Temporal Graph Benchmark for Machine Learning on Temporal Graphs"**

**Rating:** 6
**Confidence:** 3
**Clarity:** The paper is well written and underst…

**Strengths:**

Describe the strengths of the submission, considering the significance of the contribution, relevance to the broader research community, quality of the research, and ethical and social implications.

1. The authors have developed diverse and large-scale datasets, encompassing various domains, and introduced diverse evaluation tasks, including link prediction and property prediction.
2. The paper extensively tests multiple baseline models across different datasets and appropriately selects more suitable evaluation metrics. The findings reveal several intriguing and interesting results.
3. The authors provide an automated and reproducible pipeline for both link and node property prediction tasks.

**Additional Feedback:**

In negative sample sampling, why set the number of negative samples to 20？ Will this number affect the conclusion of the experiment?

**Correctness:**

The construction of dataset, experiment design, and evaluation methods appear to be correct.

**Documentation:**

The datasets, leaderboards, and details are on the corresponding website. The code is also publicly available on GitHub. There are sufficient details to support reproducibility.

**Ethics:**

There is no known ethical issue.

**Limitations:**

Yes.

**Opportunities For Improvement:**

1. The experimental setup appears to be quite limited, as it only considers evaluation under the Streaming Setting. Many baseline models may not be suitable for this setup, which could diminish the impact of this paper significantly.
2. The selection of baseline models for comparison seems insufficiently diverse, as it lacks inclusion of models like Jodie[1], TGAT[2]
3. In comparison to the findings in Reference [3], the contributions of this paper may be somewhat limited. The authors should provide a more detailed analysis and comparison with the results in Reference [25] to strengthen the paper's significance.

[1]. S. Kumar, X. Zhang, and J. Leskovec. Predicting dynamic embedding trajectory in temporal interaction networks. In Proceedings of the 25th ACM SIGKDD International Conference on Knowledge Discovery & Data Mining, 2019.
[2] Xu D, Ruan C, Korpeoglu E, et al. Inductive representation learning on temporal graphs[J]. ICLR, 2020.
[3]. F. Poursafaei, A. Huang, K. Pelrine, and R. Rabbany. Towards better evaluation for dynamic link prediction. In Thirty-sixth Conference on Neural Information Processing Systems Datasets and Benchmarks Track.

**Relation To Prior Work:**

Although the authors discuss previous works, the discussion lacks sufficient depth and detail. It is essential for the paper to provide a more comprehensive and in-depth review of the related literature.

**Summary And Contributions:**

The paper introduces Temporal Graph Benchmark (TGB), offering diverse and challenging datasets for evaluating machine learning models on temporal graphs. TGB spans large-scale, long-duration datasets in various domains, including social, trade, and transportation networks. Evaluation protocols based on real-world use-cases are provided for node and edge-level prediction tasks. Common models' performance varies significantly across datasets, and simple methods outperform existing models in dynamic node property prediction tasks. The findings encourage further research in temporal graph analysis. TGB provides an automated machine learning pipeline for reproducible research and is open to community feedback, maintenance, and updates.

---

> ### Author Response · Authors · 2023-08-17
> **Author Response to Reviewer yfpt**
>
> We thank the reviewer for their time and valuable feedback.
>
> **Q1: only considers evaluation under the Streaming Setting. Many baseline models may not be suitable for this setup**
>
> **A:** In this work, we focus on the streaming setting on continuous time dynamic graphs (CTDG) which is widely adopted in the literature. All of the methods we compared in this work are applicable in the streaming setting and we also included more methods including NAT and TGAT in the revision. For more discussion on method details,  see *Appendix D*. Future CTDG methods can also submit to the TGB leaderboard.
>
> **Q2: lacks inclusion of models like JODIE, TGAT**
>
> **A:** We have added TGAT in *Section 5.1* and started running experiments for JODIE. However, with the JODIE implementation from DyGLib, we observe a GPU memory error after 19 epochs. We are looking to fix this error in the future. We refer the reviewer to our joint reply to all reviewers for more discussion on additional baselines.
>
> **Q3: provide a more detailed analysis and comparison with the results in Reference [1]**
>
> **A:** In [1], the authors showed the limitation in current evaluation settings for dynamic link prediction and added six small datasets with less than five million edges. In this work, we focus on building a realistic and comprehensive set of datasets and tasks for benchmarking temporal graph methods. We contributed seven novel datasets with up to 73 million edges, including both edge and node level tasks. We discussed this comparison in the revised *Section 2*. We will continue to improve TGB based on community feedback and add additional datasets in the future.
>
> [1] F. Poursafaei, A. Huang, K. Pelrine, and R. Rabbany. Towards better evaluation for dynamic link prediction. In Thirty-sixth Conference on Neural Information Processing Systems Datasets and Benchmarks Track.
>
> **Q4: more comprehensive and in-depth review**
>
> **A:** In the revised *Section 2*, we added discussions comparing our work with prior TG libraries and benchmarks (including an added comparison with the DGraph [1] dataset). In *Appendix D*, we discussed details about continuous time and discrete time dynamic graphs and added in depth discussion of all methods compared in this work. We would also be glad to add any additional references suggested by the reviewer.
>
> [1] X. Huang, Y. Yang, Y. Wang, C. Wang, Z. Zhang, J. Xu, and L. Chen. DGraph: A large-scale financial dataset for graph anomaly detection. In Thirty-sixth Conference on Neural Information Processing Systems Datasets and Benchmarks Track, 2022.
>
> **Q5: why set the number of negative samples to 20? Will this number affect the conclusion of the experiment?**
>
> **A:** We selected 20 negative samples as a trade off between inference time and evaluation completeness. On medium and large tgbl datasets, inference time on the test set (with 20 negative samples) can take hours even for fast methods such as TGN and DyRep (see *Appendix J*). On small datasets i.e. tgbl-wiki and tgbl-review, the number of negative samples can be increased, which we did in the revision (see *Section 5.1*). Comparing these new results with the ones using 20 negative samples on the tgbl-wiki dataset in the table below, we find out that, as expected, many methods have a performance drop when the number of negative samples increases. For more discussion and results on tgbl-review see *Appendix H*.
>
> | Method  |   Test MRR (20 samples) |  Test MRR (all samples) |
> |----------|:-------------:|:-------------:|
> | DyRep     |   0.366   |   0.050   |
> | TGN        |   0.721   |   0.396    |
> | CAWN     |  0.791   |  0.711     |
> | TCL          |  0.712   |  0.207     |
> | GraphMixer     |  0.701   |  0.118     |
> | $EdgeBank_{tw}$     |  0.641    |  0.571     |
> | $EdgeBank_{\infty}$     |  0.538    |  0.495    |

---

> > ### Author Response · Authors · 2023-08-28
> > **Follow up discussion**
> >
> > We thank the reviewer for their insightful comments and suggestions.
> > We would greatly appreciate your input on whether we have addressed your questions and concerns, as the author-reviewer discussion period is ending soon.

---

### Official Review · Reviewer_j6Gx · 2023-07-26
**A timely benchmark, but insufficient discussion and experiments.**

**Rating:** 7
**Confidence:** 4
**Correctness:** Yes
**Clarity:** Yes

**Strengths:**

(S1) Timely Benchmark Datasets

Temporal graph mining research has received considerable attention recently and addresses a variety of practical problems. Therefore, the provision of new datasets and the evaluation of existing methods for such research is timely and important.

(S2) Large and Diverse Datasets

The authors provide datasets from a variety of domains that are significantly larger than existing benchmark datasets.

**Additional Feedback:**

Please refer to (Opportunities For Improvement)

**Documentation:**

Yes

**Limitations:**

The paper discusses the limitations and their potential negative social impact.

**Opportunities For Improvement:**

(W1) Insufficient Discussion of the Node Property Prediction Task

W1-1: The introduction lacks a comprehensive discussion on the importance and necessity of the newly proposed node property prediction task by the authors.

W1-2: It is unclear why the authors claim that the node property prediction task is a node-level task. The considered tasks focus on predicting the weights of links rather than specific property of individual nodes.

(W2) The Necessity of Some Datasets

The utility and necessity of some datasets remain unclear. For example, regarding tgbn-flight, significance and applications of predicting flights on a specific day are not well-explained. Similarly, regarding tgbn-genre, why do we need to predict the relatively trivial genre as opposed to what music a user will listen to?

(W3) Inadequate Evaluation

W3-1: Considering the significance of a temporal graph method's inductive ability for dynamic graphs, it becomes imperative for a benchmark to facilitate separate experiments in both the inductive and transductive settings, as typically performed in previous research. Unfortunately, the current evaluation lacks this essential support.

W3-2: Why not use CAWN, TCL, and GraphMixer for the task of dynamic node property prediction? It would be better to provide a clear justification for their exclusion.

**Relation To Prior Work:**

Yes,

**Summary And Contributions:**

This paper introduces a novel Temporal Graph Benchmark (TGB) comprising large and diverse datasets across various domains, including interaction, rating, transaction, social, trade, and traffic. The authors also propose two evaluation protocols: dynamic link property prediction and dynamic node property prediction. Moreover, they conduct extensive evaluations, comparing four temporal graph methods with several baselines, and offer a public leaderboard for reference.

---

> ### Author Response · Authors · 2023-08-16
> **Author Response to Reviewer j6Gx**
>
> We thank the reviewer for their time and valuable feedback.
>
> **W1-1: the introduction lacks a comprehensive discussion on the importance and necessity of the newly proposed node property prediction task by the authors.**
>
> **A:** thanks for raising this point, we have added more discussion on the importance of the node property prediction task in the Introduction. This task is motivated by applications in recommendation systems where the goal is to predict how the user preference towards items shifts over time.
>
> **W1-2: it is unclear why the authors claim that the node property prediction task is a node-level task. The considered tasks focus on predicting the weights of links rather than specific property of individual nodes.**
>
> **A:** Node property prediction and link prediction are closely connected tasks, however, their main distinction lies in the way these tasks are defined. For link prediction, we are looking at the likelihood of positive links being higher than negative non-existent links. However, in node property prediction, we consider the affinity of a node towards different items as its property. We choose to represent this as a node level task as it is feasible to compute full logits, i.e., the number of classes are relatively small.  Moreover, the task is difficult, as shown in *Section 5.2*: strong TG methods which perform well on link prediction are outperformed by simple baselines in the node property prediction task.
>
> **W2: the necessity of some datasets**
>
> **A:** We believe both tgbl-flight and tgbn-genre are useful and practical datasets and we added more task discussions in *Section 4*. For tgbl-flight, predicting the flight on a future date is important for traffic forecasting, especially when predicting disruptions such as cancellation and delays. In addition, predicting global flight networks is also important for studying the spread of disease such as COVID-19 to new regions, as seen in [1,2].
> For tgbn-genre, predicting preference towards music genres is important for music recommendation systems that aim to provide personalized (or node based) recommendations. In this dataset, the genre preference of users often shift significantly over time, thus increasing task difficulty.
> In *Section 5.2*, we also showed that on this dataset, state-of-the-art TG models have low NDCG performance when compared to simple baselines.
>
> [1] I. I. Bogoch, A. Watts, A. Thomas-Bachli, C. Huber, M. U. Kraemer, and K. Khan. Potential for global spread of a novel coronavirus from china. Journal of travel medicine, 27(2):taaa011, 369 2020.
>
> [2] X. Ding, S. Huang, A. Leung, and R. Rabbany. Incorporating dynamic flight network in seir to model mobility between populations. Applied Network Science, 6(1):1–24, 2021.
>
>
> **W3-1: facilitate separate experiments in both the inductive and transductive settings**
>
> **A:** We refer the reviewer to our joint response to all reviewers. We have added an additional experiment for transductive vs. inductive setting in *Appendix I* and also shown in the following table (for test set of tgbl-wiki). We also showed the amount of inductive nodes in TGB datasets in *Table 5* and believe that both transductive and inductive reasoning is required for a method to perform well on TGB leaderboards.
>
> | Method  |   Transductive MRR |  Inductive MRR |
> |----------|:-------------:|:-------------:|
> | DyRep     |   0.040   |   0.053   |
> | TGN        |   0.325   |   0.268    |
> | CAWN     |  0.720   |  0.709     |
> | TCL          |  0.194   |  0.182     |
> | GraphMixer     |  0.122    |  0.107     |
> | **TGAT**         |  0.155    |  0.152     |
> | **NAT**            |  **0.763**    |  **0.717**   |
> | $EdgeBank_{tw}$     |  0.575    |  0.551     |
> | $EdgeBank_{\infty}$     |  0.481    |  0.562    |
>
>
> **W3-2: why not use CAWN, TCL, and GraphMixer for the task of dynamic node property prediction**
>
> **A:** CAWN, TCL and GraphMixer are designed specifically for link prediction and only link prediction experiments are conducted in the original papers. Adopting these methods for node-level tasks is not a trivial endeavor and would demand substantial modifications to their original implementations. For integration of these and other methods in the future, we have provided an automated and accessible framework that facilitates the evaluation of TG methods on TGB datasets. Moreover, we are committed to maintaining TGB, continuously welcoming novel submissions from the community to update the leaderboards.

---

> > ### Comment · Reviewer_j6Gx · 2023-08-24
> > **Response to Authors**
> >
> > I thank the authors for responding to my concerns in detail, and I believe the authors have addressed most of my concerns.
> >
> > However, I still have a concern about the reviewer's response to the node property prediction task.
> > I believe that the downstream tasks of graph mining (not specific applications) should be defined and named from a more general perspective rather than from a specific application perspective.
> > The definition of this task seems to hold only for bipartite graphs that have a direction, i.e., in cases where the interactions are determined by a specific type of node (e.g., users in recommendation). However, this view does not seem to hold for general unipartite graphs. In many graph mining domains dealing with unipartite graphs, this task would be considered a link property, where the properties of both nodes within an edge are considered, rather than the property of a particular node.
> >
> > While my concern may seem trivial to the authors, I would say that this definition is very important for a benchmark paper that will be referenced by related researchers in the future. As an individual researcher with a strong interest in temporal graphs, I still appreciate the authors' work and hope that this paper will be published and benefit a wide range of researchers. I will gladly raise my rating if the authors' additional arguments and thoughts on these concerns are reasonable and valid.

---

> > > ### Author Response · Authors · 2023-08-25
> > > **Follow up Response to Reviewer j6Gx**
> > >
> > > We thank the reviewer for the detailed discussion and appreciation of our work. We are also glad to hear that we have addressed most of your concerns. We agree with the reviewer that the node property prediction task should be defined as a general category and not from a specific application perspective. We have revised *Section 3.2* to reflect this in the updated manuscript. First, we have renamed the current node level task included in TGB appropriately as *node affinity prediction* as it focuses on node affinity towards items over time. This should avoid any confusion between this task and the common node classification or node regression tasks. Next, we have redefined *dynamic node property prediction* to be a general task category for node level tasks on temporal graphs including subtasks such as node affinity prediction, node classification and node regression. The distinction here is also made in other parts of the manuscript for consistency.
> > >
> > > When we started developing TGB, there was a lack of large scale temporal graph datasets with node labels, therefore we mainly focused on the node affinity prediction task as a first step towards node level tasks on large datasets. We have now identified the DGraph dataset [1] as a candidate to be included into TGB in the near future (kindly suggested by Reviewer 87xo).  The DGraph dataset contains binary anomaly labels for 1.2 million nodes while also being a unipartite user graph. Therefore, we hope the addition of node classification for this dataset in the final version of our paper would address the reviewer’s concern.  We will also continue to incorporate new datasets into TGB in the future to further increase the task diversity.
> > >
> > > We hope that the terminology modification (namely having *dynamic node property prediction* refer to a general set of tasks on nodes, and *node affinity prediction* referring to the current task we introduced in the tgbn datasets) addressed your concern and we would be happy to answer further questions.
> > >
> > > [1] Huang, Xuanwen, Yang Yang, Yang Wang, Chunping Wang, Zhisheng Zhang, Jiarong Xu, Lei Chen, and Michalis Vazirgiannis. "Dgraph: A large-scale financial dataset for graph anomaly detection." Advances in Neural Information Processing Systems 35 (2022): 22765-22777.

---

> > > > ### Comment · Reviewer_j6Gx · 2023-08-25
> > > > **Thanks for your detailed response**
> > > >
> > > > I appreciate the addition of a general definition for the node property prediction task.
> > > > In addition, considering the node affinity prediction as a subtask of the node property prediction task makes a lot of sense to me.
> > > > The authors have now addressed all of my concerns, and I believe this work will benefit the community. Therefore, I will raise my rating from 5 to 7.
> > > >
> > > > FYI, it is my understanding that the additional page of the paper for revisions is only accepted by one page, but your paper currently has a total of 11 pages. Please see the track chairs' rebuttal guidelines.

---

> > > > > ### Author Response · Authors · 2023-08-28
> > > > > **Follow up discussion**
> > > > >
> > > > > We appreciate the reviewer's input in highlighting the page limit criteria.
> > > > > We have made necessary formatting adjustments while ensuring the paper's content remains intact.
> > > > > We also thank the reviewer for their support and consideration.

---

### Author Response · Authors · 2023-08-16
**General Response to All Reviewers**

We thank the reviewers for their insightful and valuable feedback. We are glad to see that all reviewers (*j6Gx*,*yfpt*,*WDxN*,*87xo*,*12NE*) unanimously agreed that our novel datasets are diverse and large scale. Reviewer *j6Gx* and *WDxN* acknowledged the importance of our research and reviewer *87xo* stated we have made “a clear contribution to the dynamic graph learning community.” Reviewer *yfpt* and *12NE* also agreed that our experiments revealed interesting findings.

We have revised the manuscript based on suggestions from reviewers (changes are marked by blue in the revised manuscript). TGB is a community driven project, thus we will continue to incorporate feedback from the reviewers and the community. In addition to the detailed replies to each reviewer, we address some common points raised by the reviewers below and mention updates made in the revision.

**P1. Inclusion of Transductive vs. Inductive Setting**

Based on [1,2], inductive nodes are nodes that are not observed in the training set. Based on this definition, we consider the nodes in the validation and test set as either transductive or inductive nodes. We report the inductive node statistics for tgbl datasets in the table below (also in *Table 5* in the paper). We observe that three out of five datasets exceed the 10% inductive nodes ratio set in previous work [1,2]. Thus, achieving a high MRR on TGB datasets requires models to excel at both transductive and inductive reasoning. We also added an additional experiment for tgbl-wiki in *Appendix I* where separate MRR are reported for the transductive and inductive nodes.

| Dataset   |   Val. Inductive Node Ratio |  Test Inductive Node Ratio |
|----------|:-------------:|:-------------:|
| `tgbl-wiki`     |   0.257   |    0.308   |
| `tgbl-review` |   0.024   |   0.027    |
| `tgbl-coin`     |  0.112    |  0.174     |
| `tgbl-comment`     |  0.474    |  0.562     |
| `tgbl-flight`     |  0.031    |  0.045     |

[1] Y. Wang, Y.-Y. Chang, Y. Liu, J. Leskovec, and P. Li. Inductive representation learning in
temporal networks via causal anonymous walks. In International Conference on Learning
Representations

[2] D. Xu, C. Ruan, E. Korpeoglu, S. Kumar, and K. Achan. Inductive representation learning on temporal graphs. In International Conference on Learning Representations, 2020.

**P2. Additional Baselines**

We have added two additional methods, NAT and TGAT, to the link property prediction experiments. Currently a total of eight TG methods are compared. In the table below, we compare the performance of the added methods (bolded) on tgbl-wiki dataset. For more results see the revised *Section 5.1*. We will continue to adapt other methods to TGB datasets. One of the objectives of TGB is to invite the community to develop novel models for temporal graph learning and then submit to the TGB leaderboard. Recently, we received submissions from the community and updated our leaderboard accordingly.

| Method   |   Val. MRR |  Test MRR |
|----------|:-------------:|:-------------:|
| DyRep     |   0.072   |    0.050   |
| TGN        |   0.435   |   0.396    |
| CAWN     |  0.743    |  0.711     |
| TCL          |  0.198    |  0.207     |
| GraphMixer     |  0.113    |  0.118     |
| **TGAT**         |  0.131    |  0.141     |
| **NAT**            |  **0.773**    |  **0.749**     |
| $EdgeBank_{tw}$     |  0.600    |  0.571     |
| $EdgeBank_{\infty}$     |  0.527    |  0.495    |

**P3. Training Time and GPU Usage Comparisons**

We have added *Figure 5* reporting the total training and validation time for each method in *Section 5.1*. We have also reported the GPU usage information in *Appendix F*. For training and validation time, we also observe that there can be orders of magnitude difference between TG methods, thus suggesting that scalability as an important research direction.

**P4. Added Novel Large Scale `tgbn-token` Dataset**

We have added a novel dataset: tgbn-token, with around 73 million edges for the node property prediction task. Experimental results on this dataset are added to *Section 5.2*. We will continue to add novel datasets to TGB in the future.

**P5. Updates Based on Community Feedback**

From our public code repository and website, we have received positive and constructive community feedback. Based on the suggestions, we have updated the tgbl-wiki and tgbl-review datasets (the small scale link property prediction datasets) to include more negative samples (see revised *Table 2*). Moreover, we have corrected an error in the node property prediction metric computation and reported revised results in *Table 4*. It should be noted that the conclusion of the experiments remain unchanged: simple baselines outperform current models in node property prediction. We will continue to monitor and integrate community feedback.

---

### Decision · Program_Chairs · 2023-09-22

**Decision:**

Accept (Poster)

**Comment:**

The paper introduces a new benchmark: Temporal Graph Benchmark (TGB). It is comprehensive with datasets from multiple domains and both node and edge-level tasks. The automatic pipeline is easy to use which will great ease the research in this domain. The paper is recommended for an acceptance.